

# Muscle size explains low passive skeletal muscle force in heart failure patients

Fausto Antonio Panizzolo[1,2], Andrew J. Maiorana[3,4], Louise H. Naylor[2], Lawrence G. Dembo[5], David G. Lloyd[6], Daniel J. Green[2,7] and Jonas Rubenson[2,8]

[1] John A. Paulson School of Engineering and Applied Sciences, Harvard University, Cambridge, MA, United States
[2] The School of Sport Science, Exercise and Health, The University of Western Australia, Crawley, WA, Australia
[3] Advanced Heart Failure and Cardiac Transplant Service, Royal Perth Hospital, Perth, WA, Australia
[4] School of Physiotherapy and Exercise Science, Curtin University, Perth, WA, Australia
[5] Envision Medical Imaging, Perth, WA, Australia
[6] Centre for Musculoskeletal Research, Griffith Health Institute, Griffith University, Gold Coast, QLD, Australia
[7] Research Institute for Sport and Exercise Science, Liverpool John Moores University, Liverpool, United Kingdom
[8] Biomechanics Laboratory, Department of Kinesiology, The Pennsylvania State University, University Park, PA, United States

Corresponding author
Fausto Antonio Panizzolo, fpanizzolo@seas.harvard.edu

## ABSTRACT

**Background.** Alterations in skeletal muscle function and architecture have been linked to the compromised exercise capacity characterizing chronic heart failure (CHF). However, how passive skeletal muscle force is affected in CHF is not clear. Understanding passive force characteristics in CHF can help further elucidate the extent to which altered contractile properties and/or architecture might affect muscle and locomotor function. Therefore, the aim of this study was to investigate passive force in a single muscle for which non-invasive measures of muscle size and estimates of fiber force are possible, the soleus (SOL), both in CHF patients and age- and physical activity-matched control participants.
**Methods.** Passive SOL muscle force and size were obtained by means of a novel approach combining experimental data (dynamometry, electromyography, ultrasound imaging) with a musculoskeletal model.
**Results.** We found reduced passive SOL forces (∼30%) (at the same relative levels of muscle stretch) in CHF vs. healthy individuals. This difference was eliminated when force was normalized by physiological cross sectional area, indicating that reduced force output may be most strongly associated with muscle size. Nevertheless, passive force was significantly higher in CHF at a given absolute muscle length (non length-normalized) and likely explained by the shorter muscle slack lengths and optimal muscle lengths measured in CHF compared to the control participants. This later factor may lead to altered performance of the SOL in functional tasks such gait.
**Discussion.** These findings suggest introducing exercise rehabilitation targeting muscle hypertrophy and, specifically for the calf muscles, exercise that promotes muscle lengthening.

## INTRODUCTION

Growing evidence suggests that architectural and functional deficiencies (e.g., strength) in skeletal muscle contribute to the limited ability to perform daily tasks and the overall poor exercise tolerance that characterizes chronic heart failure (CHF) and to the progression of the disease (*Green et al., 2016*). For example, it is apparent that patients with CHF have a reduction in muscle size (*Mancini et al., 1992*; *Minotti et al., 1993*; *Anker et al., 1999*; *Fülster et al., 2013*) and strength (as determined by net joint moments) in the lower limbs (*Magnusson et al., 1994*; *Chua et al., 1995*; *Harrington et al., 1997*; *Sunnerhagen et al., 1998*; *Toth et al., 2006*; *Toth et al., 2010*; *Panizzolo et al., 2015*) compared to healthy age-matched individuals; these reductions are also related to reductions in aerobic exercise capacity ($\dot{V}O_2$ peak) (*Volterrani et al., 1994*; *Harrington et al., 1997*; *Panizzolo et al., 2015*). It is still not clear, however, if the reduction in muscle function and aerobic capacity are associated primarily with reduced muscle size that is known to occur in CHF (*Mancini et al., 1992*; *Fülster et al., 2013*; *Panizzolo et al., 2015*) or if size-independent characteristics is an important determinant. Indeed, several studies that have measured both voluntary strength and muscle size in the quadriceps suggest that muscle size alone does not account for the loss of strength (*Harrington et al., 1997*; *Toth et al., 2006*; *Toth et al., 2010*). Resolving whether muscle size or other size-independent muscle properties are more closely linked to muscle function can prove important for guiding rehabilitation strategies in CHF.

To this extent, measurements of passive muscle forces and how they are related to muscle architecture can provide important information for understanding the mechanisms behind the alterations in skeletal muscle function associated with CHF. In particular, they can shed further light on whether skeletal muscle deficits at a whole muscle level are related primarily to reductions in muscle size without introducing variability arising from voluntary and/or twitch contractions (*Pincivero et al., 2000*; *Oskouei et al., 2003*). Passive forces are also functionally relevant as they influence normal (*Whittington et al., 2008*) and pathological (*Geertsen et al., 2015*) gait mechanics.

Our understanding of how passive skeletal muscle force is affected in CHF is currently unclear. Passive forces in cardiac muscle are altered in CHF (*Van der Velden, 2011*), as well as in diaphragm skeletal muscle (*Van Hees et al., 2010*). Surprisingly, as far as we are aware, only one study (*Hees et al., 2010*) has investigated passive forces in appendicular skeletal muscle in CHF and it has been conducted in a mouse model. This study reported unaltered passive forces in the soleus (SOL) muscle of CHF-affected mice, compared to a control group, when taking into consideration muscle size.

The SOL has been identified as a primary muscle in which tissue loss occurs in CHF (*Panizzolo et al., 2015*; *Green et al., 2016*) and its size is strongly correlated with the reduced exercise capacity present in CHF (*Panizzolo et al., 2015*) (more so than the gastrocnemius synergist). The SOL is also functionally relevant as it has been identified as the main source of mechanical work during gait (*McGowan, Kram & Neptune, 2009*). Furthermore, the SOL permits an estimation of passive force in a single muscle (*Rubenson et al., 2012*; *Tian et al., 2012*) and thus is a muscle of choice for muscle-specific analysis.

**Table 1** Anthropometric characteristics of the participants involved in the study. Data are means ± SD. Criteria for exclusion for the CHF group included severe renal (creatinine > 250 mmol/l or eGFR < 30 ml/min/1.73 m$^2$) or hepatic (bilirubin > 50 mmol/l) dysfunction or unexplained anemia (hemoglobin < 100 g/l) or thrombocytopenia (platelets < 100 * 109/l). Participants with the following contra-indications to exercise were excluded: unstable angina or exercise induced ischemia at low exercise levels (less than three metabolic equivalent units), severe aortic stenosis, severe mitral or aortic regurgitation, or hypertrophic cardiomyopathy.

| Group | Age (yr) | Height (cm) | Weight (kg) |
|---|---|---|---|
| Control | 62.7 ± 5.6 | 173.3 ± 6.1 | 69.7 ± 8.5 |
| CHF | 63.5 ± 10.9 | 168.2 ± 9.6 | 67.9 ± 14.8 |

Therefore, the aim of this study was to investigate the passive forces in the SOL muscle of CHF patients and age- and physical activity-matched control participants, including their relationship to muscle architecture (physiological cross sectional area (PCSA), muscle length, pennation angle). We hypothesized that there would be a reduction in passive force in CHF patients, compared to a healthy population. We further hypothesized that passive force would be similar after normalizing for the muscle PCSA, thus attributing alterations in passive force to muscle size.

## MATERIALS AND METHODS

### Participants

Patients with CHF and age- and physical activity-matched control participants who were free from other musculoskeletal disorders and lower limb musculoskeletal injuries were recruited for this study. The CHF group included 12 participants (seven men, five women) in the class II–IV of the New York Heart Association (NYHA) classification with an ejection fraction of 30.5 ± 9.6%. (For anthropometric characteristics and exclusion criteria see Table 1.) The control group was composed of 12 healthy participants recruited from the local community (eight men, four women). The CHF group underwent regular exercise activity 2–3 times per week for ∼1 h per session (treadmill walking and resistance weight training) as part of their standard patient care. The control participants were selected from those reporting similar levels of weekly exercise, assessed by means of a fitness questionnaire (*Godin & Shepard, 1985*). All participants read and signed an informed consent prior to participating in the study and all of the procedures were approved by the Human Research Ethics Committee at The University of Western Australia (approval ID: RA/4/1/2533) and Royal Perth Hospital (approval ID: 2011/019).

### Passive force estimates

The procedures used to estimate passive SOL forces were similar to those adopted previously (*Rubenson et al., 2012*), with the exception that passive force was measured during continuous joint rotation. Passive moments were recorded with the participants sitting upright with their right foot and ankle positioned in a dynamometer (Biodex M3, Shirley, NY, USA) and with the knee positioned at 120° of flexion (0° knee fully extended) to mitigate the force contribution of the gastrocnemius muscles (*Maganaris, 2001*). The

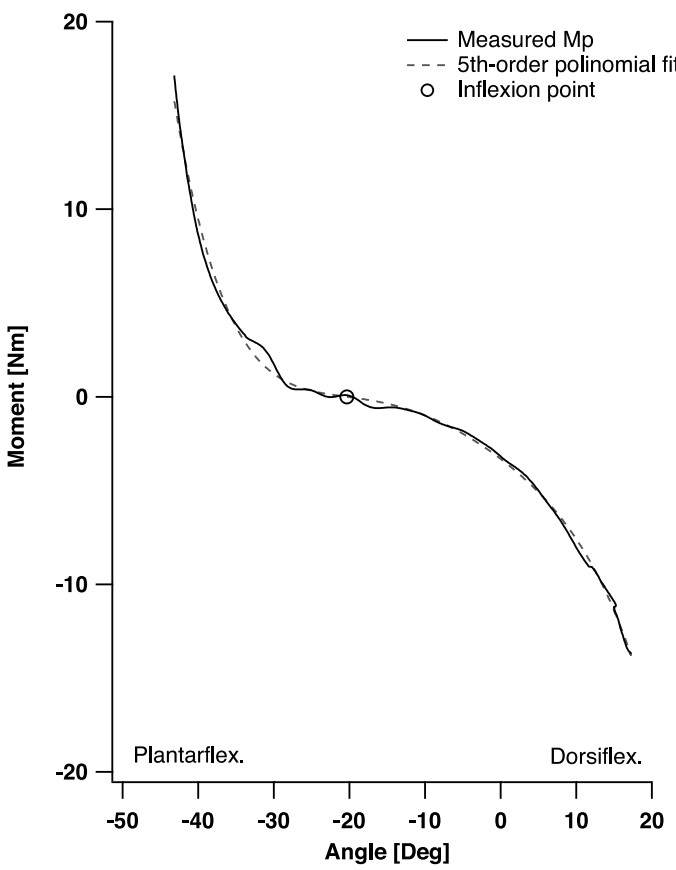

**Figure 1 Net passive ankle joint moment-angle relationship for one representative participant.** The measured moment is displayed with solid black line, the 5th-order polynomial fit with dashed black line, the inflexion point where net dorsiflexion and plantarflexion moment converge on zero is displayed with a black circle (○).

net passive ankle joint moment ($M_p$) was computed by subtracting the moment generated by the Biodex rig and the weight of the foot (*Rubenson et al., 2012*); the weight of the foot was expressed as a percentage of body mass. The $M_p$ over a joint's range of motion passes through zero at an angle that approximates where passive muscle forces reach zero (*Silder et al., 2007*) (Fig. 1). Moment data recorded by the dynamometer were filtered using 4th-order zero-lag 2 Hz low-pass Butterworth filter (MATLAB; The MathWorks Inc., USA). To detect the inflexion point in $M_p$ where net dorsiflexion and plantarflexion moment converge on zero we first fitted the joint angle vs. $M_p$ data with a 5th-order polynomial and subsequently computed the first order derivative of this function (MATLAB; The MathWorks Inc., USA) (Fig. 1).

Contribution from synergist muscles are minimal at the joint postures adopted (*Maganaris, 2001*; *Silder et al., 2007*; *Rubenson et al., 2012*). This was confirmed from OpenSim musculoskeletal models simulations (*Delp et al., 2007*) further indicating that passive force from synergist muscles were minimal at the recorded knee and ankle postures. The method described above does not account for passive moments arising from joint

articulations and skin, but these are minimal compared to the passive moments arising from passive force in the Achilles tendon (*Costa et al., 2006*).

Electromyography (EMG) from the tibialis anterior (TA), the medial and lateral gastrocnemius muscles (MG, LG, respectively) and the SOL were recorded during the trials (Noraxon wireless system, Scotsdale, AZ, USA, 2,000 Hz) to ensure the muscles crossing the ankle remained inactive. For each trial, real-time root-mean-square (RMS) waves of the muscles' activity were computed from the EMG signals (incorporating DC offset; Spike2 V7 software; Cambridge Electronic Design, Cambridge, UK) (*Rubenson et al., 2012*). Soleus fascicle lengths and pennation angle were recorded using dynamic B-mode ultrasound (Telemed, EchoBlaster 128, Lithuania; 25 Hz capture rate; 7.5 MHz 60 mm linear array probe) following the placement and image analysis procedures outlined previously (*Rubenson et al., 2012*; *Panizzolo et al., 2013*). Simultaneous measurements of ankle joint flexion/extension angles were made using a portable 3D motion capture system (100 Hz; Optitrack, Corvallis, Oregon, US). The net joint moment, EMG, ultrasound images and joint angles were recorded synchronously (Micro1401-3; Cambridge Electronic Design, Cambridge, UK; 2,000 Hz) as the ankle was cycled through its full range of motion (the most plantarflexed and most dorsiflexed position tolerated by the participant) at a constant speed of 5°/s over three consecutive cycles.

Three initial warm-up cycles were performed prior the recording of any measurements. The SOL passive force ($F_{p_{SOL}}$) was computed continuously throughout the joint range of motion as the joint underwent dorsiflexion. Passive force was calculated as per (*Rubenson et al., 2012*) using the following equation:

$$F_{p_{SOL}} = \frac{M_p}{r * \cos\theta} \qquad (1)$$

where $r$ represents the Achilles moment arm data and $\theta$ the SOL pennation angle, measured according to *Panizzolo et al. (2015)*.

Participant-specific Achilles moment arm data were established experimentally on a separate testing day, following the method described previously in *Manal, Cowder & Buchanan (2010)*. In this method B-mode ultrasound (Telemed, Echo Blaster 128, Lithuania) was used to capture Achilles tendon images in the sagittal plane from the participants while their foot was cycled passively at an angular velocity of 5°/s across its range of motion in a Biodex dynamometer (M3, Biodex, Shirley, NY, USA). The ultrasound probe (7.5 MHz, 60 mm field of view, linear array probe, 50 Hz capture rate) was placed longitudinally over the Achilles tendon, so that the ankle joint was included in the field of view of the probe, using a stand-off gel pad (Aquaflex, Parker, NJ, USA). Simultaneously, the trajectories of two retro-reflective markers mounted on the ultrasound probe were recorded by means of a 3D motion capture system (100 Hz; Optitrack, Corvallis, Oregon, US). Additional anatomical landmarks (first metatarsal, calcaneus, medial malleoli and knee medial condyle) were tracked to calculate the ankle flexion/extension joint angle. A 2D customized graphical interface was developed in Matlab to display both the ultrasound images and the ultrasound probe and the medial malleolus markers in the same coordinate system. The line of action of the Achilles tendon was digitized in this common coordinate

system and the moment arm was computed as the perpendicular distance between the midline of action of the tendon and the medial malleolus, which was used as an estimate of the ankle joint center. This procedure was performed at 10 ankle joint angles that spanned the joint's range of motion. A 10-point moment arm-joint angle curve was obtained for each participant by using a polynomial fit of the moment arm-joint angle data.

We defined the fascicle slack length ($L_{slack}$) as the length where passive SOL forces are first generated. $L_{slack}$ was estimated as the point where the net passive dorsiflexion and plantarflexion moments converge on zero (Fig. 1). The fascicle length at the maximum tolerated dorsiflexion angle was defined as the maximal fascicle length ($L_{max}$). Absolute and normalized passive SOL force–length (F–L) curves were established for each participant. Absolute passive F–L curves used the measured $F_{pSOL}$ in Newtons and fascicle lengths ($L$) in mm. Normalized passive F–L curves were created from absolute passive F–L curves by dividing each participant $F_{pSOL}$ by their SOL PCSA and by dividing $L$ by $L_{slack}$ (this normalized length is referred as $L_{norm}$ from hereinafter). The PCSA was determined from underwater 3D ultrasound scans which allowed to obtain information relative to SOL muscle volume (Telemed, EchoBlaster 128, Lithuania; Stradwin, Medical Imaging Research Group, Cambridge University Engineering Department, UK) following (*Panizzolo et al., 2015*). To enable the comparison of absolute $F_{pSOL}$ between groups, $F_{pSOL}$ was determined at a percent fascicle stretch of 0%, 20%, 40%, 60%, 80% and 100% of the maximum fascicle stretch, where percent fascicle stretch was defined as $((L - L_{slack}) \div (L_{max} - L_{slack})) * 100$. The same procedure was done to compare passive moment data over both angle and muscle length ranges. Passive fascicle stiffness was computed for each participant as the slope of the absolute F–L curves between $L_{slack}$ and 40% stretch ($k_1$) and between 60%–100% stretch ($k_2$). In order to compare the normalized passive F–L curves we evaluated the normalized $F_{pSOL}$ at a set of $L_{norm}$ between 1.0 and 1.4 (i.e., strain of 0–40%) using intervals of 0.05. A peak $L_{norm}$ was set to 1.4 as this represented the average maximum $L_{norm}$ that the participants achieved at their end range of ankle dorsiflexion. The normalized $F_{pSOL}$ was computed for each individual for the interval described above by fitting the normalized $F_{pSOL}$ and $L_{norm}$ data using a 1st-order exponential equation (*Gollapudi & Lin, 2009*) with subject-specific constants. In some circumstances where the set range exceeded the experimental $L_{norm}$, the normalized $F_{pSOL}$ values were extrapolated from the exponential equation. Stiffness was computed between $L_{norm}$ of 1.0 and 1.2 ($k_{1norm}$) and 1.2 and 1.4 ($k_{2norm}$).

## Active forces estimates

As an ancillary comparison of the muscle lengths, we also analyzed peak active muscle forces at different ankle angles (and thus muscle lengths) to generate an active force–length relationship. It has previously been shown experimentally, both in the human SOL muscle (*Rubenson et al., 2012*) and in non-human muscle (*Azizi & Roberts, 2010*) that optimal muscle lengths ($L_0$; lengths where peak active isometric forces are generated) correspond closely with $L_{slack}$. Because of the importance of $L_{slack}$ in our analyses of length-dependent passive muscle force and muscle stiffness we chose to also assess $L_0$ as an additional test for differences in fascicle lengths between groups. The main purpose of performing the active force–length curve for the SOL muscle was thus to improve our assessment of

length-dependent passive force and muscle stiffness that relies on length normalization, rather than insights into active force production *per se*.

The protocol used in this study to obtain predictions of moments and force generated by the SOL (as well as the moments and force generated by synergist muscles and by the co-contraction of dorsiflexor muscles) expands on the procedures established in *Rubenson et al. (2012)*. It uses a combination of experimental net moment measurements from dynamometry, ultrasound fascicle imaging, electromyography and a scaled participant-specific musculoskeletal model in OpenSim 2.0.2 (*Delp et al., 2007*). Predictions were performed with the knee in a flexed position (>120°) and over a range of ankle angles from ∼−20° dorsiflexion to 30° plantarflexion (the ankle range of motion varied between individuals ranging from individual maximum dorsiflexion of −30° to individual maximal plantarflexion of nearly 50°). The muscle length that corresponded with the maximal peak active force was designated as $L_0$.

First, a generic OpenSim lower-limb model (*Arnold et al., 2010*) was scaled using each participant's joint axes and centers determined via motion capture data (8-camera VICON MX motion capture system, 100 Hz; Oxford Metrics, UK) from participants in a standing posture as well as dynamic joint motions (*Besier et al., 2003*). From these trials, an inverse kinematics algorithm was run on the position of 26 retroreflective spherical markers placed on anatomical landmarks and on functionally determined joint centers (*Besier et al., 2003*), that minimized the distance between the OpenSim model markers and the retroreflective and the functionally determined markers. These participants-specific models were positioned to match the participant's optically recorded ankle and knee joint posture and in turn used to predict $M_{dorsi}$ and $M_{Syn}$ (see below).

The moment generated by the plantarflexors ($M_{plant}$) during the maximal voluntary isometric plantarflexion contractions ($MVC_{plant}$) was calculated as:

$$M_{plant} = M_{peak} - \triangle M_p + M_{dorsi} \tag{2}$$

where $M_{peak}$ is the peak net ankle joint moment (calculated as the difference between the Biodex recorded moment during $MVC_{plant}$ and the moment at rest), $\triangle M_p$ represents the difference in the estimated passive SOL moment during the $MVC_{plant}$ and the passive SOL moment at rest prior to the contraction, and $M_{dorsi}$ is the moment generated by the co-contraction of the dorsiflexors muscles.

$\triangle M_p$ was calculated as:

$$\triangle M_p = \left( F_{p_{SOL}}^{contr} * \cos\theta^{contr} * r^{contr} \right) - \left( F_{p_{SOL}}^{rest} * \cos\theta^{rest} * r^{rest} \right) \tag{3}$$

where $F_{p_{SOL}}$ was obtained for both the fascicle length at the $MVC_{plant}$ and the fascicle length during the rest period just prior to contraction using a linear interpolation of the passive F–L relationship (rest and contr superscripts designate rest or $MVC_{plant}$, respectively). $r^{contr}$ was estimated by increasing the value predicted from the experimental Achilles moment arm- joint angle equation (described above) by 20% to take in account the increase in moment arm distance reported during $MVC_{plant}$ with respect to length at rest (*Maganaris, Baltzopoulos & Sargeant , 1998*).

The $M_{\text{dorsi}}$ was predicted by the participant-specific OpenSim model. First, the OpenSim maximal isometric forces of all the dorsiflexors (tibialis anterior, extensor digitorum longus, extensor hallucis longus, peroneus tertius) were adjusted by the same percentage increase or decrease so that the predicted model's peak isometric dorsiflexion moment at 100% activation ($MVC_{\text{dorsi}}$) matched that of the participant's experimental maximum $M_{\text{dorsi}}$ recorded in the Biodex dynamometer at 10° plantarflexion, the angle that corresponds approximately to optimal dorsiflexion moments (*Silder et al., 2007*). The $MVC_{\text{dorsi}}$ were performed only at this joint angle to reduce the total numbers of contractions performed and time spent in the experimental protocol by each participant. This was an important consideration because of the general high fatigability of CHF patients. In this procedure, the OpenSim model was positioned to match the participant's optically recorded ankle and knee joint posture. In subsequent measurements of $MVC_{\text{plant}}$ the $M_{\text{dorsi}}$ was predicted by the OpenSim model by prescribing an activation to all of the dorsiflexors equal to the ratio of the TA's peak EMG (linear envelope) during the $MVC_{\text{plant}}$ to its peak EMG (linear envelope) from the $MVC_{\text{dorsi}}$ trial; i.e., this assumed the same activation level for all dorsiflexors.

To take into account the contribution of synergist muscles we predicted the relative percentage contribution of each plantarflexors muscle to the total plantarflexor moment in OpenSim ($M_{\text{Syn}}$) by prescribing the recorded ankle and knee angles and 100% activation of all plantarflexor muscles (peroneus longus, peroneus brevis, flexor hallucis, tibialis posterior, flexor digitorum, MG, LG and SOL). The percent contribution of the OpenSim SOL to the total predicted moment was applied to the experimental $MVC_{\text{plant}}$ to define the moment generated by the participant's SOL ($M_{a_{\text{SOL}}}$). Lastly, peak voluntary active SOL force production ($F_{a_{\text{SOL}}}$) was calculated as:

$$F_{a_{\text{SOL}}} = \frac{M_{a_{\text{SOL}}}}{r^{\text{contr}} * \cos\theta^{\text{contr}}}. \tag{4}$$

These active force trials were performed only by the participants that were able to tolerate a prolonged protocol ($n = 7$ and $n = 8$, for control and CHF participants, respectively).

## Statistical analysis

Differences in the absolute (non-normalized) passive moment–angle, moment-length and F–L curves were assessed by testing if $F_{p_{\text{SOL}}}$ were different between groups (CHF and control), and if joint angles at which the passive forces occurred and/or fascicle lengths were affected in the CHF group, by using a two-way (CHF/control) repeated measures (0%, 20%, 40%, 60%, 80% and 100% of angular excursion or muscle stretch, respectively) ANOVA, with Bonferroni *post hoc* tests. Similar two-way repeated measures ANOVAs were also performed on the normalized F–L curves using the $L_{\text{norm}}$ set range (1.0–1.4). A two-tailed unpaired Student's $t$-test with significance level of $p < 0.05$ was used to determine significant differences in the $L_{\text{slack}}$, $L_{\text{max}}$, the maximal fascicle stretch, and $L_0$, as well as in the passive fascicle stiffness ($k_1, k_2, k_{1\text{norm}}$ and $k_{2\text{norm}}$) and in the PCSA between the groups. Finally, we performed a linear regression analyses between peak $F_{p_{\text{SOL}}}$ at $L_{\text{norm}}$ of 1.4 and PCSA and between muscle volume across groups. Statistical analysis was performed in SPSS (IBM, Statistics 21, USA).

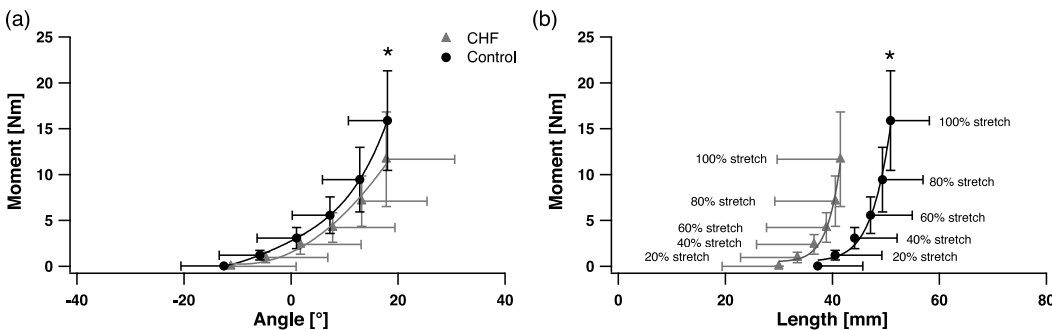

**Figure 2** Passive moment-ankle angle relationship (evaluated at 0%, 20%, 40%, 60% and 100% of the ankle's maximal dorsiflexion) (A) and passive ankle moment-soleus length relationship (B). The chronic heart failure (CHF) group is displayed in grey triangles (▲), and control group in black circles (●). Average curves are displayed ± S.D. ∗ designates a statistical difference in passive ankle moment between groups (ANOVA main effect; $p < 0.05$; CHF vs. control).

## RESULTS

No main effect of group was found in the joint angle between the CHF and control groups ($p = 0.42$) (Fig. 2A). A main effect of group on net passive ankle joint moment was found ($p = 0.01$) with lower passive moment in the CHF group compared to the control group at relative levels of angular excursion and fascicle stretch, although no statistically significant interaction effect was found ($p = 0.40$) between group and joint angle (Figs. 2A–2B).

A main effect of group on absolute $F_{p_{SOL}}$ (N) was found ($p = 0.03$) with lower absolute $F_{p_{SOL}}$ in the CHF group compared to the control group at relative levels of fascicle stretch, although no statistically significant interaction effect was found ($p = 0.11$) between group and level of stretch. No differences were found in $k_1$ and $k_2$ between the groups ($p = 0.32$; ES = 0.51 and $p = 0.85$; ES = 0.09) (Fig. 3A), with stiffness exhibiting high variability. The $L_{max}$ was significantly shorter in the CHF group compared to the control group ($p = 0.046$; ES = 0.96), although no statistically significant differences were found in $L_{slack}$ ($p = 0.11$; ES = 0.76) and in the maximal fascicle stretch ($L_{max} - L_{slack}$) ($p = 0.34$; ES = 0.44) (Table 2) or maximal fascicle strain ($p = 0.70$; ES = 0.09).

A significantly smaller SOL PCSA was found in the CHF with respect to the control group ($p = 0.02$; ES = 1.25) (Table 2). No main effect was found in the PCSA-normalized $F_{p_{SOL}}$ (N cm$^{-2}$) between the CHF and control groups when using the $L_{norm}$ strain range of 1.0–1.4 ($p = 0.46$) (Fig. 3B), nor was there an interaction effect between the PCSA-normalized $F_{p_{SOL}}$ and normalized lengths ($p = 0.52$). Normalized passive fascicle stiffness ($k_{1norm}$ and $k_{2norm}$) were likewise variable and also not significantly different between the groups ($p = 0.42$; ES = 0.44 and $p = 0.54$; ES = 0.33) (Fig. 3B).

$L_0$ determined from the active force–length data was significantly shorter (∼22%) in the CHF group compared to the control group ($p = 0.04$; ES = 0.96) (Table 2). The maximal $F_{a_{SOL}}$ and corresponding $L_0$ occurred at approximately 10° dorsiflexion. The $F_{a_{SOL}}$ at both shorter and longer fascicle lengths relative to $L_0$ decreased, characteristic of the muscle force–length relationship (Fig. 4). $L_0$ was not significantly different from $L_{slack}$ in either the control or CHF groups ($p = 0.33$ and $p = 0.39$, respectively; Table 2).

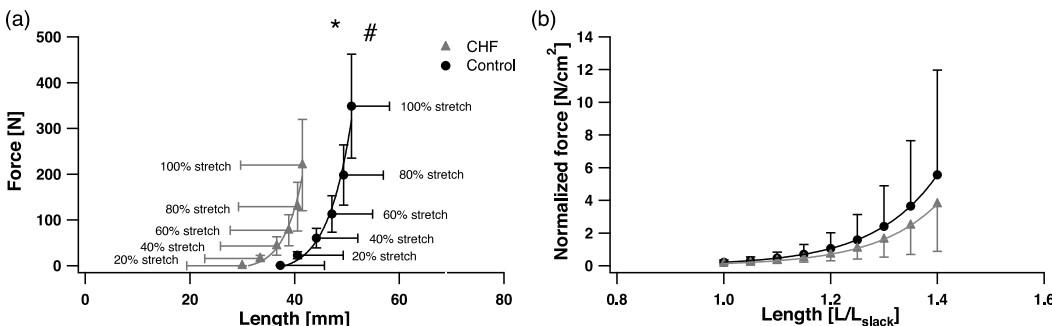

**Figure 3** **Soleus passive force-length (F-L) relationship (A) and passive F-L relationship normalized by individual PCSA and $L_{slack}$ (B).** Passive F-L is evaluated at 0%, 20%, 40%, 60% and 100% of the muscle's maximal stretch (A), passive F-L is evaluated at $L_{norm}$ between 1.0–1.4 (B). The chronic heart failure (CHF) group is displayed in grey triangles (▲), and control group in black circles (●). Average curves are displayed $\pm$ S.D. $*$ designates a statistical difference in passive force between groups (ANOVA main effect; $p < 0.05$; CHF vs. control). # designates a significant difference in maximal passive fascicle length ($L_{max}$) between groups ($p < 0.05$; CHF vs. control).

**Table 2** **Muscle parameters.** Data are means $\pm$ SD.

| SOL muscle parameter | CHF | Control |
|---|---|---|
| $L_{slack}$ (mm) | $30.0 \pm 10.6$ | $37.3 \pm 8.4$ |
| $L_{max}$ (mm) | $41.5 \pm 11.8^{*}$ | $50.9 \pm 7.3$ |
| $L_{max} - L_{slack}$ (mm) | $11.5 \pm 5.2$ | $13.6 \pm 4.2$ |
| $L_0$ (mm) | $26.4 \pm 6.4^{*}$ | $33.7 \pm 8.4$ |
| Max strain ($L_{norm}$) | $1.4 \pm 0.2$ | $1.4 \pm 0.2$ |
| PCSA (cm$^2$) | $65.0 \pm 13.0^{*}$ | $91.0 \pm 20.5$ |
| Max $F_{pSOL}$ (N) | $220.0 \pm 99.6^{*}$ | $348.6 \pm 113.4$ |
| $k_1$ (N/mm) | $6.9 \pm 2.9$ | $12.0 \pm 13.8$ |
| $k_2$ (N/mm) | $75.1 \pm 78.9$ | $82.6 \pm 85.7$ |
| $k_{1norm}$ | $2.8 \pm 1.7$ | $4.1 \pm 4.1$ |
| $k_{2norm}$ | $18.0 \pm 14.8$ | $26.3 \pm 32.2$ |

**Notes.**
$^{*}$Indicates a significant difference ($p < 0.05$).

A significant correlation was found between peak $F_{pSOL}$ at $L_{norm}$ of 1.4 and muscle volume ($p < 0.01$; $r = 0.76$) while no significant correlation was reported between peak $F_{pSOL}$ at $L_{norm}$ of 1.4 and PCSA ($p = 0.06$; $r = 0.46$).

## DISCUSSION

The present study provides, to the best of our knowledge, the first estimate of *in vivo* passive human skeletal muscle force–length properties in CHF. As predicted, higher absolute $M_p$ and $F_{pSOL}$ were produced in the control group for a given amount of muscle stretch (Figs. 2 and 3). However, and also in agreement with our hypothesis, passive force was not different after normalizing by muscle PCSA, nor is passive muscle stiffness affected. These results indicate that muscle size rather than intrinsic muscle properties is a major factor influencing

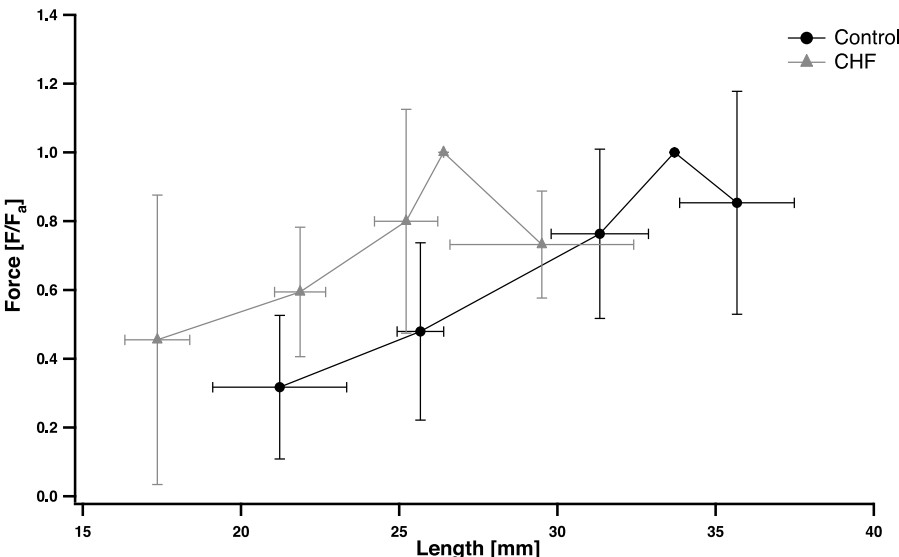

**Figure 4  Normalized soleus peak voluntary force production and fascicle length.** Peak voluntary force production is normalized by the individual's maximal peak voluntary force production ($F_{a_{SOL}}$) while fascicle length is expressed in mm. Data points (mean $\pm$ S.D.) include measurements taken at joint angles greater and less than 10° dorsiflexion. Muscle lengths were grouped in four clusters equally spaced along the fascicle length range. A data point at ($L_0, 1$) is included for reference purposes only. The chronic heart failure (CHF) group is displayed by grey triangles (▲), and control group by black circles (●).

passive forces in CHF SOL muscle. This was further supported by the correlation between peak $F_{p_{SOL}}$ and muscle size across both the CHF and control groups. This finding stands in contrast to previous work reporting stiffer cardiac muscle due to alterations in the titin structure (*Wu, 2002*) or decreased passive force of the diaphragm, due to titin loss (*Hees et al., 2010*) in CHF. On the other hand, our results do corroborate data from passive skeletal muscle properties in the mouse SOL, in which passive forces from CHF-affected animals were likewise not altered after normalizing to muscle cross sectional area (*Hees et al., 2010*).

It was surprising, however, that for a given absolute muscle length, passive force was significantly higher in CHF SOL compared to the control group. This unexpected finding stems from the fact that over the same ankle range of motion the passive muscle lengths are shorter in CHF patients, in particular at maximal stretch (Figs. 2 and 3). The result is that for the same absolute muscle length (above $L_{\text{slack}}$) the CHF muscle has undergone greater strain, thus generating greater force in titin and other passive load bearing muscle components. Previous experimental studies (*Azizi & Roberts, 2010*; *Winters et al., 2011*; *Rubenson et al., 2012*) have shown agreement between the onset of passive force generation ($L_{\text{slack}}$) and $L_0$ (optimal length for active force production). The estimate of $L_0$ in the present study was similar to $L_{\text{slack}}$ for both groups but significantly ($p < 0.05$) shorter in the CHF group (Table 2). The shorter $L_{\text{slack}}$ and $L_0$ in CHF patients may indicate that the SOL has undergone a loss of in-series sarcomere numbers, a contributing factor to the reduced muscle size (*Panizzolo et al., 2015*). It was also surprising that, despite their shorter muscle fascicles, CHF patients underwent the same ankle range of motion and a similar SOL muscle strain across this range of motion (Fig. 2 and Table 2). Therefore, the 'effective'

stiffness of the muscle, the amount of force resulting at the maximal stretch of the muscle (as indicated by $k1$ and $k2$), are similar between groups despite the passive force at the same absolute muscle length being substantially greater in CHF patients. The passive moments at equivalent ankle angle excursions (indicative of the ankle's effective angular stiffness) are likewise similar between CHF and control groups (Fig. 2). This is true except for a moderately higher moment, and absolute force, in the control group at the participant's peak dorsiflexion angle (Figs. 2 and 3) although these angles are rarely achieved during normal movement tasks. The Achilles moment arms were similar between the control and CHF group suggesting that greater Achilles strain might explain the similarity in joint and muscle excursions. This is partially supported by the smaller tendon cross sectional area reported in CHF (*Panizzolo et al., 2015*).

## Functional implications

Our results are consistent with the observation that muscle size dictates functional deficits in CHF (*Magnusson et al., 1994*). Exercise that promotes hypertrophy should therefore be a focus for restoring functional capacity in leg muscles. Exercise prescription for CHF is becoming commonplace, but programs that include specifically designed lower limb resistance training might be especially promising (*Maiorana et al., 2000*).

Our results also offer insight into the gait mechanics of CHF patients (*Panizzolo et al., 2014*). The combination of the shorter SOL muscle fascicles in CHF patients and their greater dorsiflexion during mid-stance of gait (*Panizzolo et al., 2014*) may cause significantly greater SOL strain. This might lead to the muscle operating on to the descending limb of the F–L curve where large passive forces develop (*Rassier, MacIntosh & Herzog, 1999*; *Rubenson et al., 2012*). In this scenario CHF patients would rely more on their passive forces to support the plantarflexion moment during walking, which has the benefit of reducing metabolically expensive active force development. This may help explain why CHF patients rely proportionately more on their ankle for powering walking as speed and metabolic demand increases (*Panizzolo et al., 2014*). However, whilst metabolically advantageous, this mechanism might lead to greater lengthening-induced muscle damage. Alternatively, a larger dorsiflexion during the stance phase could be explained by a higher tendon strain, without affecting the SOL strain itself. The muscle's F–L operating range and its interplay with the Achilles tendon function depend on multiple factors, including tendon stiffness and a detailed understanding will require further *in vivo* analyses to clarify the underlying mechanism.

## Limitations

Some limitations of the present study need to be acknowledged. First, in the measurement of $M_p$ used to calculate passive forces estimates, some participants displayed an inflection point (where net dorsiflexion and plantarflexion moment converge on zero, Fig. 1) slightly above or below zero moment (<1.5 Nm or ~7% of the peak passive moment). This can occur if the weight of the leg transmits a small moment about the Biodex axis (i.e., small misalignment of ankle center of rotation) or if the moment predicted from the weight of the foot has small errors. In these cases the passive moment data was corrected for the offset. Second, the method used to calculate the Achilles moment arm data (*Manal, Cowder &*

*Buchanan, 2010*), assumed the position of the ankle joint center coincides with the marker placed on the medial malleolus. This could have resulted in a potential misalignment of the ankle joint center, which might have affected the estimation of the moment arm measurements. Investigations have also shown that muscle fascia structure can act as a pathway for myofascial force transmission (*Purslow, 2010*), thus making more difficult to completely isolate fascicle force production at single muscle level. Nevertheless, this factor has been reported to be relatively small in intact muscles (*Maas & Sandercock, 2010*) and most likely did not significantly impacted our findings. Lastly, the heterogeneity of the CHF group needs to be acknowledged as it might have influenced some of the findings. The difficulty associated with enrolling large numbers of CHF participants to undertake prolonged biomechanical tests prevented us from controlling for variables such as body mass, composition, stature, or sex. Nevertheless, we tried to mitigate this problem by recruiting closely matched age and physical activity-level control participants.

## CONCLUSION

This work suggests that a primary factor leading to lower passive forces in the SOL is likely a reduction in muscle size. However, shorter muscle fascicles in CHF results in greater passive forces for a given absolute muscle length, and might be linked to changes in CHF gait (*Panizzolo et al., 2014*). Exercise that promotes calf muscle hypertrophy and serial sarcomerogenesis may prove particularly beneficial in CHF patients.

## ACKNOWLEDGEMENTS

The authors would like to acknowledge Ms. Kirsty McDonald and Ms. Maddison Jones for their help during data collection, and all of the participants who volunteered for this study.

### Funding

This work was supported by a Grant-in-Aid (G09P 4469) from the National Heart Foundation of Australia to JR, DJG, AJM and DGL, and a thesis dissertation grant from the International Society of Biomechanics to FAP. The funders had no role in study design, data collection and analysis, decision to publish, or preparation of the manuscript.

### Grant Disclosures

The following grant information was disclosed by the authors:
National Heart Foundation: G09P 4469.
International Society of Biomechanics.

### Competing Interests

Lawrence G. Dembo is an employee of Envision Medical Imaging. The authors declare there are no competing interests.

## Author Contributions

- Fausto Antonio Panizzolo conceived and designed the experiments, performed the experiments, analyzed the data, wrote the paper, prepared figures and/or tables, reviewed drafts of the paper.
- Andrew J. Maiorana conceived and designed the experiments, performed the experiments, contributed reagents/materials/analysis tools, reviewed drafts of the paper.
- Louise H. Naylor conceived and designed the experiments, performed the experiments, reviewed drafts of the paper.
- Lawrence G. Dembo conceived and designed the experiments, contributed reagents/materials/analysis tools, reviewed drafts of the paper.
- David G. Lloyd and Daniel J. Green conceived and designed the experiments, analyzed the data, contributed reagents/materials/analysis tools, reviewed drafts of the paper.
- Jonas Rubenson conceived and designed the experiments, performed the experiments, analyzed the data, contributed reagents/materials/analysis tools, wrote the paper, prepared figures and/or tables, reviewed drafts of the paper.

## Human Ethics

The following information was supplied relating to ethical approvals (i.e., approving body and any reference numbers):

All participants read and signed an informed consent prior to participating in the study and all of the procedures were approved by the Human Research Ethics Committee at The University of Western Australia (approval ID: RA/4/1/2533) and Royal Perth Hospital (approval ID: 2011/019).

## Data Availability

The raw data has been supplied as Data S1.

## Supplemental Information

Supplemental information for this article can be found online at http://dx.doi.org/10.7717/peerj.2447#supplemental-information.

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
