# Peer review of "Muscle size explains low passive skeletal muscle force in heart failure patients"

_PeerJ, doi:10.7717/peerj.2447_

## Round 0.1 · original submission · Minor Revisions

· Academic Editor

Minor Revisions

As you will notice, the reviewers had general positive comments, though they did raise a number of points that when addressed should improve the clarity of exposition. Note also that both reviewers had some difficulties with the interpretation of figures, particularly figures 1 and 2. I also appreciated the suggestion of reviewer 1 to add a graph displaying stiffness. Another relevant issue is the explicit mentioning of passive forces. Please carefully review the comments on the presentations of some of the statistical results, and please consider introducing multi-linear regressions! Some good feedback was also given in stylistic choices (long paragraphs and so forth). Please address those as well.

Looking forward to read the revised manuscript!

·

Basic reporting

The article is well written and general clear. Figures are appropriate and clear although should be better referenced in the text - parts A and B often show very different things and the results could be better followed if specific parts were referenced.

An additional figure (bar chart?) could be included to show stiffness values. These seem central to the paper so should be more prominently displayed and easier to assess. This may also allow different values (using different normalization - see validity of findings section) to be displayed and functional significance assessed.

Sufficient data (including raw data) is presented to test the stated hypothesis.

Experimental design

The experiment appears to have been well conducted. The muscle in question has been isolated. EMG was used to assess activity levels during passive trials. Care has been taken to match activity levels - often a confounding factor in such studies. Both men and women are used in this study, and given the high levels of variability in the data, it may be worth considering whether this has an effect on the findings.
Ethical standards appear to have been met.

Some care needs to be taken throughout about the language used to describe what the study addresses. The language in the discussion and conclusion is appropriate as it states that changes in passive force are a result of changes in muscle size and the goes on to speculate about what these changes might mean for locomotor performance in CHF patients. However, in other places there is more of an implication that the study is assessing whether reduction in size or quality underpin the ability of muscle to generate force actively in CHF patients. This is particularly true in the title and introduction. The title uses ‘quality’. Quality is typically used to refer to the ability of muscle to produce active force and so should be removed. The introduction talks about the loss of voluntary strength in CHF patients and using passive force to understand whether this kind of decline in active is a result of a reduction in muscle size or muscle quality. However, the study does not test this and there are many factors which would complicate assessing active force production from passive properties. It should therefore be made explicit throughout that passive forces are being assessed and any comment on the implications of this left to the discussion.

A better description of the range of angles rotated though maybe helpful, plantar flexion and dorsiflexion could be annotated on figure 1?

In the methods, pennation angles are measured. But these are never described. And passive force is corrected using a pennation angle. This correction makes sense for active force as force is produced along the length of a fiber. But makes a little less sense for passive force.

There is a section on calculating active force-lengths curves. Active force in the soleus is estimated over a range of lengths. However, this is not well integrated into either the rationale for the study or the discussion of findings. If this section is to remain, it should be better incorporated into the study.

Validity of the findings

The data collected are largely appropriate to the conclusions drawn.

The data as a whole shows very high variability. This is a common issue in human studies with limited subjects and it appears that care has been taken to minimize this where possible. However, this variability should be acknowledged, particularly when showing a lack of significant differences between groups.

All of the pertinent data measured should be presented. For example, there does not seem to be clear presentation of PCSA anywhere. This is essential as the question is about muscle size.

I am a little confused about some of the data. It is unclear to me how joints can be rotated through the same range of angles with smaller fiber length changes – this is what figures 2A and B appear to show? Additional stretch occurs in the tendon? This should be addressed somewhere. More explicit presentation of the rotation conditions and morphological parameters in the 2 groups early on in the results may help this.

The authors should carefully consider normalization throughout, and the implications of this should be better explained. For example, calculating stiffness from normalized forces and lengths gives the most information about muscle properties. However, this ignores the higher effective stiffness experienced in CHF patients. In some cases it may be better to calculate a variety of parameters and discuss the implications of them. Significant differences in L0 are shown, however, the normalized plot obscures these.

This is also true of the stats where non-normalized values are compared at 0, 20, 40%….. of maximal angular excursion and stretch. You should be very explicit about what these comparisons mean. For example, if angular excursions are similar between the groups this is more like an absolute comparison. However, if maximal muscle stretch differs between the groups, using these categories is effectively comparing normalized lengths. These things don’t necessarily need to be changed but it would be beneficial to the reader to understand what is being compared. If possible, it may also be beneficial to use more sophisticated statistical analysis. Linear or generalized linear regressions could be used in R to assess data without using the % categories

Some of the stats terminology is a little confusing. For example It is not clear to me how an interaction between group and moment would be found. Interactions occur between 2 independent variables (e.g. group and sex) not a dependent and an independent variable (e.g. group and moment) (line 284). This use of interaction should be addressed throughout the results.

Reasonable conclusions are drawn from the data and some potential functional implications discussed.

·

Basic reporting

I have the following minor recommendations regarding the figures:
Figure 1. Please update the x- and y-axis labels to designate the sign convention. For example, in both, it seems dorsiflexion is the positive convention.

Figure 2. Do these data also correspond to the 20-100% stretch labels shown on Figure 3? If so, I would include them here as well.

Figures 2-3. I suggest arranging these panels side-by-side rather than top-to-bottom to avoid possible attempts to compare these panels across their x-axes.

Captions. In all captions, I recommend making note of the muscle being investigated (e.g., "Soleus passive force-length").

Experimental design

No Comments.

Validity of the findings

No comments.

Additional comments

This generally well-written manuscript primarily describes a comparison of passive soleus muscle forces obtained via a combination of in vivo measurement and computer simulation in people with and without chronic heart failure. In my opinion, the study is well-justified and experimentally sound. I do have several minor comments, but most are geared toward improving the clarity of the presented material.

Abstract:
L28. “Impaired skeletal muscle” is too vague.
L31. Revise: “altered contractile properties [and/or] architecture” as differentiating these roles is of primary concern.
L38 and L42. The difference between “absolute muscle length” and “equivalent levels of muscle stretch” should be clarified, perhaps by referring to the former using “relative.”
L45-46. This closing statement is not a complete sentence.

Introduction:
L54. I suggest more specifically defining/describing “deficiencies in skeletal muscle” and “limited functional capacity.”
L60. More clearly describe the relation between aerobic exercise capacity and muscle size and strength.
L62-L63. This sentence is very difficult to follow - I suspect “reduction in muscle” is a typo.
L73. This first use of “motor deficits” is overly vague.
L76. Silder et al. found little to no difference in passive contributions to net joint moments between old and young adults during walking. Is this a good example for the functional relevance of passive forces?
L85. Consider revising: “…investigate passive forces in the…. participants, including their relationship to muscle architecture…”
L88-L93. I recommend introducing the importance of the soleus muscle during functional activities and in CHF prior to stating the study’s aim.

Methods:
L107. The activity level of participants may or may not be well-matched. CHF subjects’ exercise is prescribed and detailed according to their standard patient care, but my guess is that controls simply self-reported their physical activity. I suggest revising to: “The control participants self-reported similar levels of weekly exercise.”
L113. First mention of active force measurement. Introduction suggested these would be excluded. I also recommend the authors add or move (e.g., L150-155) to this section details regarding the study protocol.
L126. I do not know what it means to fit the data by visual inspection.
L133-135. This one sentence description of the modeling efforts are unclear until more details are provided 4-pages later. Suggest reorganizing for clarity.
L136-138. This sentence should be moved to a new paragraph added to the discussion on study limitations.
L136-177. This is a huge paragraph. I recommend several shorter paragraphs with clear topic sentences.
L165-166. The word “above” could be interpreted as “over” or “proximal” and should be replaced for clarity. Where exactly along the tendon was this placed? Also, are two motion capture markers enough to co-register the ultrasound images with the medial malleoli markers? Finally, use of the medial malleoli as a surrogate for ankle joint center should be discussed as a limitation.
L172. Was the tendon’s line of action defined using its midline? If so, please state this.
L178-181. This sentence is quite difficult to follow.
L196-201. It is not clear why the authors could not simply use the measured passive forces rather than resorting here to model fits.
L205. The different ankle angles referred to here are never reported, but should be.
L221. My interpretation here is that inverse kinematics was used to drive the model to match marker trajectories measured from subjects seated in the Biodex and performing the MVCs. If this is true, I recommend stating.
L270. The “passive joint angles” outcome measure should be more clearly defined. Also, the comparison described by “were affected” is unclear.
L276. Revise: “stiffnesses”

Results:
L292. I recommend moving “maximal strain” to the next paragraph, as it is derived from the normalized curves presented in Figure 3b.
L299. This sentence largely repeats that from the methods section and can be removed.
L301. The relevance of reporting the angle at which Lo occurred is not clear.

Discussion:
L309. Revise: “However, and also in agreement with our hypothesis, passive force was not different between groups after… These results indicate that muscle size…”
L312. The inclusion of “stiffness” here is a bit misleading. Neither absolute nor normalized stiffness differed between groups. Thus, it would seem that neither muscle size nor intrinsic properties influence stiffness in CHF.
L326. Revise: “… for both groups [but] significantly…”
L327. Revise: “… patients [may indicate] that…”
L344. As an alternative, consistent with ideas proposed on L332-334, more dorsiflexion during stance could emerge due to more tendon strain, without affecting muscle strain, in people with CHF. Because of this possibility, I find the discussion of a reliance on passive muscle forces as an energy conserving mechanism in CHF highly speculative.
L353. As discussed earlier, I suggest the authors add a thoughtful discussion of study limitations.

---

## Round 0.2 · Minor Revisions

· Academic Editor

Minor Revisions

Rather than a 'number of Minor Revisions', it is in fact just one correction in figure 4 that is necessary, as proposed by the reviewer. After your resubmission, the next step should be publication.

·

Basic reporting

The article is now generally well reported.

The only change I would suggest is an alteration to figure 4. L0 is now discussed in absolute terms showing a difference between control and CHF patients and the implications of this discussed. However, the figure still shows normalized data meaning it has little relevance to the text.

The authors may wish to revise this

Experimental design

The experiment was well designed and conducted and the data now fully test the presented hypotheses.

Validity of the findings

The study presents valid findings and draws appropriate conclusions from them.

Additional comments

I find the manuscript much improved and suitable for publication. Although the authors may wish to consider how figure 4 fits with the description of the results

---

## Author Rebuttal · Round 0.2

# Reviewer 1 (Natalie Holt)

*We thank the reviewer for her thorough review and constructive criticisms. We have addressed the reviewer's concerns and we feel that the revised manuscript is improved as a result of these helpful suggestions.*

## Basic reporting

The article is well written and general clear. Figures are appropriate and clear although should be better referenced in the text - parts A and B often show very different things and the results could be better followed if specific parts were referenced.
*We thank the reviewer for this comment and we have now added figure labels as suggested.*

An additional figure (bar chart?) could be included to show stiffness values. These seem central to the paper so should be more prominently displayed and easier to assess. This may also allow different values (using different normalization - see validity of findings section) to be displayed and functional significance assessed.
*We agree with the reviewer. We realize that we did not provide the actual values for stiffness, which will be useful for comparisons. While stiffness values are important we believe that the comparison of passive force normalized by muscle size (PCSA) is a key variable. We now report further on stiffness values and note the variability in these measures. We chose to present stiffness data in Table 2.*

Sufficient data (including raw data) is presented to test the stated hypothesis.

## Experimental design

The experiment appears to have been well conducted. The muscle in question has been isolated. EMG was used to assess activity levels during passive trials. Care has been taken to match activity levels - often a confounding factor in such studies. Both men and women are used in this study, and given the high levels of variability in the data, it may be worth considering whether this has an effect on the findings.
Ethical standards appear to have been met.
*We thank the reviewer for this comment. The fact that we include both men and women has likely led to increased variability. We appreciate that our sample is heterogeneous, both with regard to sex, age and anthropomorphic data. Part of this problem stems from the difficulty in recruiting large number of CHF participants that are amenable to prolonged experiments. CHF patients are a notoriously fragile patient population with very poor stamina and frequent rehospitalization. We have made efforts to control for variability, at least in part, by recruiting age- and activity-level matched participants.*

*We have made efforts to further assess the effect of variability that mixed sex may have had on our results. We have matched participants with similar age and sex in order to perform pair-wise comparisons. This resulted in 6 pairs (not all participants were able to be matched). Our pair-wise comparisons found similar statistical findings to those of the main analysis. We have not added this to the manuscript but do discuss in further detail the limitations of our*

*sample.*

*Finally, we should note that we do feel that including both male and female participants in initial new work is important. For example, the problem of typical 'male only' studies have been highlighted recently by the American National Institutes of Health, which promotes the study of both sexes as a guiding principle for biomedicine, in particular pre-clinical work (Clayton, J.A., FASEB, 30, 000– 000 (2016).*

Some care needs to be taken throughout about the language used to describe what the study addresses. The language in the discussion and conclusion is appropriate as it states that changes in passive force are a result of changes in muscle size and the goes on to speculate about what these changes might mean for locomotor performance in CHF patients. However, in other places there is more of an implication that the study is assessing whether reduction in size or quality underpin the ability of muscle to generate force actively in CHF patients. This is particularly true in the title and introduction. The title uses 'quality'. Quality is typically used to refer to the ability of muscle to produce active force and so should be removed. The introduction talks about the loss of voluntary strength in CHF patients and using passive force to understand whether this kind of decline in active is a result of a reduction in muscle size or muscle quality. However, the study does not test this and there are many factors which would complicate assessing active force production from passive properties. It should therefore be made explicit throughout that passive forces are being assessed and any comment on the implications of this left to the discussion.

*We thank the reviewer for this comment. We agree that using the term quality can be misleading. Accordingly, we have now removed the word "quality" from the title and from the introduction. We have also made explicit throughout the manuscript that we assessed passive forces. We do retain some discussions surrounding active force and strength, however as these are important for framing the previous literature that builds the current view that skeletal muscle is an important factor limiting exercise capacity in CHF.*

A better description of the range of angles rotated though maybe helpful, plantar flexion and dorsiflexion could be annotated on figure 1?
*Thank you for this comment, we have now added the requested annotation on figure 1 and also described the angle range in the text.*

In the methods, pennation angles are measured. But these are never described. And passive force is corrected using a pennation angle. This correction makes sense for active force as force is produced along the length of a fiber. But makes a little less sense for passive force.
*We thank the reviewer for this comment, although we believe that pennation angle should be included in the calculation of passive forces since the passive forces developed in the fascicles (imaged) are greater than those in the tendon when there is a pennation angle present. This is in line with the procedures adopted also by previous studies (e.g. Rubenson et al. 2012, Tian et al. 2012) and by common musculoskeletal software such as OpenSim and SIMM when predicting passive forces and resulting passive joint moments. We agree that details of this procedure should be added to the manuscript and we added the appropriate references for this measurement.*

There is a section on calculating active force-lengths curves. Active force in the soleus is

estimated over a range of lengths. However, this is not well integrated into either the rationale for the study or the discussion of findings. If this section is to remain, it should be better incorporated into the study.

*We thank the reviewer for this suggestion and we have now incorporated our rationale for the calculation of active forces (in particular force-length properties) in the methods section, and as a result the ensuing discussion of this finding is more aligned with our main question of length-dependent passive force.*

*"As an ancillary comparison of the muscle lengths, we also analyzed peak active muscle forces at different ankle angles (and thus muscle lengths) to generate an active force-length relationship. It has previously been shown experimentally, both in the human soleus muscle (Rubenson et al., 2012) and in non-human muscle (Azizi and& Roberts, 2010) that optimal muscle lengths ($L_0$; lengths where peak active isometric forces are generated) correspond closely with $L_{slack}$. Because of the importance of $L_{slack}$ in our analyses of length-dependent passive muscle force and muscle stiffness we chose to also assess $L_0$ as an additional test for differences in fascicle lengths between groups. The main purpose of performing the active force-length curve for the soleus muscle was thus to improve our assessment of length-dependent passive force and muscle stiffness that relies on length normalization, rather than insights into active force production per se."*

## Validity of the findings

The data collected are largely appropriate to the conclusions drawn.

The data as a whole shows very high variability. This is a common issue in human studies with limited subjects and it appears that care has been taken to minimize this where possible. However, this variability should be acknowledged, particularly when showing a lack of significant differences between groups.

*Thank you for this comment. We have now added a limitation section, as suggested also by reviewer 2, that discussed the issue of variability. See also our discussion regarding the previous comment about variability resulting from including both males and females.*

All of the pertinent data measured should be presented. For example, there does not seem to be clear presentation of PCSA anywhere. This is essential as the question is about muscle size.

*We thank the reviewer for this suggestion. PCSA data are now presented in the results section and specific values are added in Table 2.*

I am a little confused about some of the data. It is unclear to me how joints can be rotated through the same range of angles with smaller fiber length changes – this is what figures 2A and B appear to show? Additional stretch occurs in the tendon? This should be addressed somewhere. More explicit presentation of the rotation conditions and morphological parameters in the 2 groups early on in the results may help this.

*The reviewer has interpreted the figures correct. In the manuscript we try to make this connection. We now state:*

*"It was also surprising that, despite their shorter muscle fascicles, CHF patients underwent the same ankle range of motion and a similar SOL muscle strain across this range of motion (Figure 2, Table 2). Therefore, the 'effective' stiffness of the muscle, the amount of force resulting at the maximal stretch of the muscle (as indicated by $k_1$ and $k_2$), are similar between*

*groups despite the passive force at the same absolute muscle length being substantially greater in CHF patients. The passive moments at equivalent ankle angle excursions (indicative of the ankle's effective angular stiffness) are likewise similar between CHF and control groups (Figure 2). This is true except for a moderately higher moment, and absolute force, in the control group at the participant's peak dorsiflexion angle (Figure 2, 3) although these angles are rarely achieved during normal movement tasks. The Achilles moment arms were similar between the control and CHF group suggesting that greater Achilles strain might explain the similarity in joint and muscle excursions. This is partially supported by the smaller tendon cross sectional area reported in CHF (Panizzolo et al., 2015)."*

The authors should carefully consider normalization throughout, and the implications of this should be better explained. For example, calculating stiffness from normalized forces and lengths gives the most information about muscle properties. However, this ignores the higher effective stiffness experienced in CHF patients. In some cases it may be better to calculate a variety of parameters and discuss the implications of them. Significant differences in L0 are shown, however, the normalized plot obscures these.

*We agree that presenting non-normalized values can help when interpreting the data and for future comparisons to other populations. Non-normalized passive forces are now summarized in the results section and were also presented in Fig. 2 and 3. $L_{slack,}$ and $L_0$ values are presented in Table 2. We now also discuss the concept of 'effective' stiffness following the good suggestion by the reviewer. We discuss that for a given absolute muscle length passive force is substantially higher in CHF. However, CHF patients never experience the same muscle lengths as the control participants, although they do undergo a similar muscle strain. We interpret effective stiffness as our measurements of $k_1$, and $k_2$ which are similar between groups, since this is the absolute force (non normalized) relative to the 'effective' maximal muscle stretch that the individual experiences in vivo over their range of motion. Also, we now discuss the ankle stiffness (ankle angular excursion vs joint moments), which likewise are similar between groups. We feel that these are important distinctions and thank the reviewer for highlighting the importance of discussing further both normalized and non-normalized data.*

This is also true of the stats where non-normalized values are compared at 0, 20, 40%….. of maximal angular excursion and stretch. You should be very explicit about what these comparisons mean. For example, if angular excursions are similar between the groups this is more like an absolute comparison. However, if maximal muscle stretch differs between the groups, using these categories is effectively comparing normalized lengths. These things don't necessarily need to be changed but it would be beneficial to the reader to understand what is being compared. If possible, it may also be beneficial to use more sophisticated statistical analysis. Linear or generalized linear regressions could be used in R to assess data without using the % categories.

*We have modified the text to better explain what the normalized and non-normalized values represent functionally and with respect to the muscle properties. We now have discussed 'effective' stiffness both with respect to the muscle as well as angular excursion and passive moments that are related to functional ('effective') ankle stiffness. We have considered regression analyses. We have performed linear regressions between non-normalized passive force (peak force) and muscle PCSA and between non-normalized peak passive*

*force and volume. These variables were chosen since they most closely address the question of whether muscle size explains passive force. These regressions show a significant correlation (r = 0.76) between muscle volume and non-normalized peak force (p < 0.01) and a correlations of r = 0.46 (p = 0.06) between PCSA. This information further strengthens the interpretation that muscle size explains peak force in the CHF group. We thank the reviewer for useful suggesting performing regression analyses. However, performing multifactorial regression analyses is problematic because the main variables would each be considered a covariate (e.g. muscle length is a covariate of muscle PCSA and muscle volume).*

Some of the stats terminology is a little confusing. For example It is not clear to me how an interaction between group and moment would be found. Interactions occur between 2 independent variables (e.g. group and sex) not a dependent and an independent variable (e.g. group and moment) (line 284). This use of interaction should be addressed throughout the results.

*We thank the reviewer for this comment. The lack of interaction effect between group and moment was a typo. This should have read no interaction effect between group and joint angle. In general though we believe the terminology used is correct. Our statistical procedures are repeated measures ANOVAs on variables of interest (e.g. passive force) between group (CHF and control) and across multiple levels of muscle lengths, muscle stretch or joint angles (repeated measures). Therefore there are two 'main effects' that the statistic is testing for, namely if there is an effect of 1) group (across all repeated measures) and 2) muscle lengths/stretch/joint angles (across both groups). It also tests for an interaction between the group and the repeated measure (muscle lengths/stretch/joint angles). An interaction effect would indicate that the relationship between the variable of interest (e.g. passive force) is affected differently by muscle lengths/stretch/joint angles in the CHF compared to control group. This statistical approach is common in repeated measures ANOVA.*

Reasonable conclusions are drawn from the data and some potential functional implications discussed.

# Reviewer 2 (Jason Franz)

*We thank the reviewer for his thorough review and constructive criticisms. We have addressed the reviewer's concerns and we feel that the revised manuscript is improved as a result of these helpful suggestions.*

## Basic reporting
I have the following minor recommendations regarding the figures:
Figure 1. Please update the x- and y-axis labels to designate the sign convention. For example, in both, it seems dorsiflexion is the positive convention.
*Thank you for this helpful comment. The angle convention has been added to the figure, as suggested also by reviewer 1.*

Figure 2. Do these data also correspond to the 20-100% stretch labels shown on Figure 3? If

so, I would include them here as well.
*Yes, they do. These labels have now been added also to Figure 2.*

Figures 2-3. I suggest arranging these panels side-by-side rather than top-to-bottom to avoid possible attempts to compare these panels across their x-axes.
*Thank you for this very useful suggestion. The panels have now been rearranged accordingly.*

Captions. In all captions, I recommend making note of the muscle being investigated (e.g., "Soleus passive force-length").
*Thank you, this has now been added in the captions.*

## Experimental design
No Comments.

## Validity of the findings
No comments.

## Comments for the Author
This generally well-written manuscript primarily describes a comparison of passive soleus muscle forces obtained via a combination of in vivo measurement and computer simulation in people with and without chronic heart failure. In my opinion, the study is well-justified and experimentally sound. I do have several minor comments, but most are geared toward improving the clarity of the presented material.
*We thank the reviewer for the positive comments regarding our study and manuscript.*

Abstract:
L28. "Impaired skeletal muscle" is too vague.
*We thank the reviewer for this comment. This has now been changed to "Alterations in skeletal muscle function and architecture".*

L31. Revise: "altered contractile properties [and/or] architecture" as differentiating these roles is of primary concern.
*Thank you, we have modified the text accordingly.*

L38 and L42. The difference between "absolute muscle length" and "equivalent levels of muscle stretch" should be clarified, perhaps by referring to the former using "relative."
*We thank you for this comment and we have now changed the terminology accordingly.*

L45-46. This closing statement is not a complete sentence.
*We thank for noticing the typo and we have now modified the sentence accordingly.*

Introduction:
L54. I suggest more specifically defining/describing "deficiencies in skeletal muscle" and "limited functional capacity."
*We thank the reviewer for this suggestion and we have now rephrased this sentence to enhance its clarity. It now reads: "Growing evidence suggests that architectural and functional*

*deficiencies (e.g. strength) in the skeletal muscle contribute to the limited ability to perform daily tasks and the overall poor exercise tolerance that characterizes chronic heart failure (CHF) and to the progression of the disease (Green et al., 2016)".*

L60. More clearly describe the relation between aerobic exercise capacity and muscle size and strength.
*We thank the reviewer for this comment, we referred to aerobic exercise capacity as $\dot{V}O_2$ peak and now we also more explicitly state that reduction in skeletal muscle functional properties are related to a reduction in aerobic capacity.*

L62-L63. This sentence is very difficult to follow - I suspect "reduction in muscle" is a typo.
*Thanks for noticing the typo, this has now been corrected.*

L73. This first use of "motor deficits" is overly vague.
*We thank the reviewer for pointing out this issue. This sentence has now been reworded for clarity; "skeletal muscle deficits at a whole muscle level".*

L76. Silder et al. found little to no difference in passive contributions to net joint moments between old and young adults during walking. Is this a good example for the functional relevance of passive forces?
*We thank the reviewer for this comment. Although we agree that the cited study did report little difference in passive contributions to net joint moments between old and young adults during walking, this is not in contradiction with our statement that "Passive forces are also functionally relevant as they influence normal gait mechanics". Nevertheless, we decided to replace this reference with another one from the same group (Whittington et al., 2007), which specifically investigates the passive-elastic mechanisms during gait without comparing different populations.*

L85. Consider revising: "…investigate passive forces in the…. participants, including their relationship to muscle architecture…"
*We thank the reviewer for the suggestion and we have now changed the wording as suggested.*

L88-L93. I recommend introducing the importance of the soleus muscle during functional activities and in CHF prior to stating the study's aim.
*We thank the reviewer for this comment and we now have rearranged this part of the introduction accordingly.*

Methods:
L107. The activity level of participants may or may not be well-matched. CHF subjects' exercise is prescribed and detailed according to their standard patient care, but my guess is that controls simply self-reported their physical activity. I suggest revising to: "The control participants self-reported similar levels of weekly exercise."
*We thank the reviewer for asking this clarification. The physical activity of both groups was assessed by means of the "Leisure-Time Exercise Questionnaire". We have added this detail in the paper and now the sentence reads as: "The control participants were selected from those reporting similar levels of weekly exercise, assessed by means of a fitness questionnaire (Godin & Shepard, 1985)".*

L113. First mention of active force measurement. Introduction suggested these would be excluded. I also recommend the authors add or move (e.g., L150-155) to this section details regarding the study protocol.

*We thank the reviewer for this comment. We prefer not to discuss active forces in the introduction and introduce their use and rationale in the methods section (See also comments from Reviewer 1 who suggests limiting focus on active force per se). Why we analyze active force and specifically a force-length curve has been clarified in the methods section (see also response to comments of reviewer 1).*

L126. I do not know what it means to fit the data by visual inspection.

*The decision to fit the data with a 5th-order polynomial (rather than with a polynomial of different order) was taken after visual inspection of the fit obtained with polynomial of different orders. We have now removed this part to avoid confusion in the readers. We have subsequently performed a more objective determination of the order number by assessing when an additional constant was not statistically different from zero. When this occurred increasing the polynomial order was not deemed to improve the fit. This resulted in $4^{th}$ and $5^{th}$-order polynomials. The difference in using a $4^{th}$- or $5^{th}$-order polynomial did not appreciably alter the prediction of $L_{slack}$*

L133-135. This one sentence description of the modeling efforts are unclear until more details are provided 4-pages later. Suggest reorganizing for clarity.

*We thank the reviewer for this suggestion. We prefer to leave the details regarding the subject-specific models in OpenSim in the section relative to the active forces estimates (as they were mainly used there). We have tried to improve the clarity of this section.*

L136-138. This sentence should be moved to a new paragraph added to the discussion on study limitations.

*We thank the reviewer for the suggestion and we have now moved this part to the discussion to a new limitations section.*

L136-177. This is a huge paragraph. I recommend several shorter paragraphs with clear topic sentences.

*We thank the reviewer for this point, we have now divided into separate paragraphs and shortened the paragraph length as suggested.*

L165-166. The word "above" could be interpreted as "over" or "proximal" and should be replaced for clarity. Where exactly along the tendon was this placed? Also, are two motion capture markers enough to co-register the ultrasound images with the medial malleoli markers? Finally, use of the medial malleoli as a surrogate for ankle joint center should be discussed as a limitation.

*We thank the reviewer for this comment, we have now replaced "above" with over and a more accurate description of the location the ultrasound probe. Also, we have now written a limitations section discussing this and the other issues mentioned by the reviewer.*

L172. Was the tendon's line of action defined using its midline? If so, please state this.

*Yes, it was. This has now been added to the paper.*

L178-181. This sentence is quite difficult to follow.
*We thank the reviewer for this suggestion and we now have reworded the sentence to enhance its clarity.*

L196-201. It is not clear why the authors could not simply use the measured passive forces rather than resorting here to model fits.
*We thank the reviewer for this comment. Measured passive forces have inherent experimental noise. By fitting each individual's data using the 1st-order exponential equation (Gollapudi & Lin, 2009) with subject-specific constants allowed us to smooth the curves and assess passive force values at the same prescribed strain values for each participant. This would have a similar effect to filtering the individual measured force-length data and using this to assemble group means. Using raw measured data would result in increased error due to signal noise. Furthermore, in some circumstances the strain range that we assessed exceeded the experimental data and in these few cases using the exponential equation allowed us to extrapolate these data.*

L205. The different ankle angles referred to here are never reported, but should be.
*We investigated the whole range of motion. The resulting joint angles were different from participant to participant due to their different flexibility. We have now reported the overall range of motion in the manuscript.*

L221. My interpretation here is that inverse kinematics was used to drive the model to match marker trajectories measured from subjects seated in the Biodex and performing the MVCs. If this is true, I recommend stating.
*We used inverse kinematics to scale a generic OpenSim lower-limb model (Arnold et al., 2010), thus creating a subject-specific model. We then positioned the models to match the participants' optically recorded ankle and knee joint posture and to predict the co-contraction and antagonist moments. We have now clarified this procedure in the text.*

L270. The "passive joint angles" outcome measure should be more clearly defined. Also, the comparison described by "were affected" is unclear.
*We thank the reviewer for this comment and we have now reworded this part to enhance its clarity. Passive joint angles were just referring to angles when the muscles are passive, but this wording is not necessary and 'passive' has been removed.*

L276. Revise: "stiffnesses"
*Thank you for noticing the typo, this has now been corrected.*

Results:
L292. I recommend moving "maximal strain" to the next paragraph, as it is derived from the normalized curves presented in Figure 3b.
*We thank the reviewer for this comment. Although we understand the reviewer's point of view, we prefer to leave this result in the previous paragraph as it is related to fascicles' stretch and so we think it might be easier for the reader to have it presented here.*

L299. This sentence largely repeats that from the methods section and can be removed.
*The sentence has now been removed accordingly.*

L301. The relevance of reporting the angle at which Lo occurred is not clear.

*$L_0$ has previously been shown to correspond well with the muscle's slack length (defined here as the length where passive forces are first generated). We decided to report the angle corresponding to $L_0$ to link muscle function with joint motion and to offer a comparison with the existing literature.*

Discussion:

L309. Revise: "However, and also in agreement with our hypothesis, passive force was not different between groups after… These results indicate that muscle size…"

*This sentence has now been revised accordingly.*

L312. The inclusion of "stiffness" here is a bit misleading. Neither absolute nor normalized stiffness differed between groups. Thus, it would seem that neither muscle size nor intrinsic properties influence stiffness in CHF.

*Agreed. We have modified the sentence accordingly.*

L326. Revise: "… for both groups [but] significantly…"

*This has now been revised accordingly.*

L327. Revise: "… patients [may indicate] that…"

*This has now been revised accordingly.*

L344. As an alternative, consistent with ideas proposed on L332-334, more dorsiflexion during stance could emerge due to more tendon strain, without affecting muscle strain, in people with CHF. Because of this possibility, I find the discussion of a reliance on passive muscle forces as an energy conserving mechanism in CHF highly speculative.

*We thank the reviewer for this interesting interpretation. We have now incorporated this hypothesis at the end of the discussion section and also highlight that the interpretations in this section remain speculative.*

L353. As discussed earlier, I suggest the authors add a thoughtful discussion of study limitations.

*We thank the reviewer for this suggestion and we have now added this section in the discussion, also in agreement with reviewer 1.*

---

## Round 0.3 · accepted · Accept

· Academic Editor

Accept

Thank you for your clear headed analysis that got to the heart of the problem (no pun), and delivered a clear and effective corroborative statement to your hypothesis. I am happy to join the reviewers in approving your manuscript, and wish you a large readership.